# Bioactive Phytochemicals with Anti-Aging and Lifespan Extending Potentials in *Caenorhabditis elegans*

**DOI:** 10.3390/molecules26237323

**Published:** 2021-12-02

**Authors:** Nkwachukwu Oziamara Okoro, Arome Solomon Odiba, Patience Ogoamaka Osadebe, Edwin Ogechukwu Omeje, Guiyan Liao, Wenxia Fang, Cheng Jin, Bin Wang

**Affiliations:** 1National Engineering Research Center for Non-Food Biorefinery, Guangxi Academy of Sciences, Nanning 530007, China; nkwachukwu.okoro.pg01008@unn.edu.ng (N.O.O.); arome.odiba@unn.edu.ng (A.S.O.); jinc@im.ac.cn (C.J.); 2College of Life Science and Technology, Guangxi University, Nanning 530007, China; wfang@gxas.cn; 3Department of Pharmaceutical and Medicinal Chemistry, University of Nigeria, Nsukka 410001, Nigeria; patience.osadebe@unn.edu.ng (P.O.O.); Edwin.omeje@unn.edu.ng (E.O.O.); 4State Key Laboratory of Non-Food Biomass and Enzyme Technology, Guangxi Academy of Sciences, Nanning 530007, China; gyliao@gxas.cn; 5Institute of Microbiology, Chinese Academy of Sciences, Beijing 100101, China

**Keywords:** aging, bioactive compounds, *Caenorhabditis elegans*, longevity pathways, lifespan

## Abstract

In the forms of either herbs or functional foods, plants and their products have attracted medicinal, culinary, and nutraceutical applications due to their abundance in bioactive phytochemicals. Human beings and other animals have employed those bioactive phytochemicals to improve health quality based on their broad potentials as antioxidant, anti-microbial, anti-carcinogenic, anti-inflammatory, neuroprotective, and anti-aging effects, amongst others. For the past decade and half, efforts to discover bioactive phytochemicals both in pure and crude forms have been intensified using the *Caenorhabditis elegans* aging model, in which various metabolic pathways in humans are highly conserved. In this review, we summarized the aging and longevity pathways that are common to *C. elegans* and humans and collated some of the bioactive phytochemicals with health benefits and lifespan extending effects that have been studied in *C. elegans*. This simple animal model is not only a perfect system for discovering bioactive compounds but is also a research shortcut for elucidating the amelioration mechanisms of aging risk factors and associated diseases.

## 1. Introduction

Various plants and their by-products have become a major area of investigation for bioactive compounds with health benefits [1,2]. These bioactive phytochemicals are chemical components (mostly secondary metabolites) that are present in relatively smaller amounts compared to macronutrients such as carbohydrates, proteins, and lipids. Depending on the specific application and benefits, these phytochemicals are classified into various categories such as medicinal, functional foods, nutraceuticals, and botanicals, some of which are closely related or almost convey the same meaning. These compounds function in other processes that are vital to plant survival, such as protection and adaptation, due to their inability to escape several potential categories of ecological and environmental (biotic and abiotic) damaging stresses. However, their benefits extend to humans and other animals that have taken advantage of these properties by sourcing such valuable components from plants for other benefits beyond the basic macronutrients that are essential for life. Most of the bioactive substances that have been discovered from food sources so far are mostly of plant origin and are consumed as fruits, legumes, vegetables, spices, and medicinal herbs [2,3]. These products are available in diverse forms, such as in fresh, raw, or processed forms. Beyond their nutritional values, these food sources contain bioactive compounds that exert anti-microbial, anti-viral, anti-carcinogenic, anti-inflammatory, antioxidants, neuroprotective, and anti-aging effects [2]. Recent research in this field has sought to discover new natural bioactive compounds in food forms that have anti-aging and lifespan extending benefits, examples of which include Silymarin and 6-shogaol [1,3,4,5,6,7].

Aging in humans is closely associated with diverse pathological changes, including cancer, cardiovascular disorders, metabolic diseases such as type II diabetes, and neurodegenerative diseases such as Alzheimer’s disease [1,2]. Conserved in all living forms, aging is a degenerative process that is characterized by a progressive deterioration of cellular components and functions, which in most cases, inevitably leads to mortality [3,4]. In developed countries, aging accounts for 90% of deaths, with about 100,000 cases per day, which makes up approximately two-thirds of deaths globally [5]. The United Nations projects that by 2050, the proportion of the global populace older that is older than 60 years of age will be closely doubled from 962 million to 2.1 billion [6]. Therefore, it is critical to gain an understanding of the molecular mechanism of the aging process along with the search for therapeutic interventions that are capable of extending lifespan and improving health span.

There are nine widely acknowledged hallmarks of aging that feature all of the major alterations of the key biological functions: loss of proteostasis, genomic instability, telomere attrition, mitochondrial dysfunction, epigenetic alterations, cellular senescence, deregulated nutrient-sensing, stem cell exhaustion, and altered intercellular communication [7]. In particular, the proteostasis network, which consists of protein synthesis, folding, secretion, trafficking, disaggregation, and degradation [7,8], is an integral part of the biological quality control systems that ensure cellular homeostasis for the survival and propagation of the organism. During aging, the decline of the proteostasis network contributes to the development of proteotoxicity related disorders such as Parkinson’s disease, Huntington’s disease, Alzheimer’s disease, and Amyotrophic lateral sclerosis [1,9]. There are two prominent concepts or theories as to how adverse changes occur in aging at the molecular level [10]. The programmed theory proposes aging as a genetically programmed chronic process, whereas the damage theory puts emphasis on the gradual and cumulative damage to cells and organs derived from internal and external factors [11]. The damage theory largely stems from the free radical theory that primarily focuses on the generation of reactive oxygen species (ROS) as metabolic by-products and their accumulative damage to biomolecules [10,11,12,13] such as DNA/RNA, proteins, and lipids. The antidotes to this fundamental challenge are the antioxidants that provide a counter mechanism to keep the balance of ROS production in check. On one hand, it secures all of the essential ROS-dependent reactions, and on the other, it removes excessive ROS and prevents the undesirable species from being generated, thereby delaying aging-associated changes and increasing longevity [4,11,12,14]. Meanwhile, the search for bioactive compounds (from natural sources as well as synthetic) possessing antioxidant properties has been greatly intensified, which has so far brought to light several candidates with interesting potential in enhancing lifespan and health span.

Many critical aspects of the human aging process have been studied in model organisms, providing us great insight into the individual elements and more importantly the operating and underlying regulatory mechanisms. Amongst these organisms, the *Caenorhabditis elegans* model has been greatly employed for the discovery of longevity pathways as well as new anti-aging compounds with lifespan extending properties [15,16,17,18]. Here, we briefly introduce *C. elegans* as an important aging model, highlighting the prominent pathways or factors that are involved in aging process, and discussed the recent discovery of lifespan-extending compounds, plant sources as well as their anti-aging activities in the *C. elegans* system.

## 2. *C. elegans* as a Model for Aging Research

Model organisms have become a crucial part of biomedical research, tackling fundamental biological and medical questions that would otherwise be impossible to study in humans due to the cost, system complexity, and ethical issues [19,20,21]. Any biological system in which aging occurs has the potential of being a model for studying aging. However, the choice of the model is largely dependent on the specific questions to be answered as well as the amenability of the model. *C. elegans*, a multicellular organism sharing 60–80% similarity with humans at the genomic level, has emerged as an outstanding model for aging research [22,23]. Featured by a highly conserved aging signaling network, *C. elegans* has a relatively short life cycle and low maintenance and propagation costs [24,25]. More so, sophisticated genetic techniques and manipulations, such as RNAi, CRISPR-cas9, and the auxin-inducible degradation (AID) system, are all applicable to *C. elegans* for transgenesis as well as forward and reverse genetic screening [26]. The transparent body of *C. elegans* also serves an ideal system for real-time live imaging of fluorescence-tagged proteins in the whole animal [27,28,29]. Despite the abounding advantages with using *C. elegans* in aging research, it is not without limitations. On the downside, the advantages that are associated with the small size of this organism, such as ease of handling, can also be a substantial disadvantage, as obtaining a large amount of the same generation is limiting as well as labor intensive. Likewise, in spite of the >60% genetic conservation with humans, *C. elegans* is indeed a lower invertebrate and is significantly distant from mammals evolutionarily, biochemically, and physiologically [30,31,32].

A series of pioneering works established the biochemical pathways that are associated with aging and longevity in *C. elegans*, which appear to be conserved through evolution [31,32,33,34,35,36]. Klass isolated eight mutant strains with a remarkable increase in lifespan and correlated this phenomenon with the restriction of caloric intake [33]. Further studies led to the identification of *age-1*, the first gene linked to lifespan extension [37,38]. Similarly, mutations in *daf-2* exhibited more than a double-fold increase in the lifespan through regulating the activity of *daf-16* [36]. These discoveries set up the foundation for using the *C. elegans* model to understand the aging process and to seek opportunities for lifespan extension. More work was inspired, collectively leading to the appreciation of the complex network underpinning the aging process. Meanwhile, this model has also been explored in other dimensions, ranging from assessing the environmental factors for aging (hermetic treatments and caloric restriction), studying the age-related diseases, population and evolutionary studies, and screening of drugs with potential lifespan-extending properties [13,39,40,41,42].

## 3. Signaling Pathways and Environmental Factors Related to Aging

Over 70 genes have been implicated in the pathways regulating the lifespan of *C. elegans* [38], with a high likelihood of further expansion on the current number. These genes are involved in the nutrient-sensing signaling pathways, including Target of Rapamycin (mTOR) signaling, AMP-activated protein kinase (AMPK)-dependent signaling, sirtuins, and insulin/IGF-1 signaling (IIS). Other implicated pathways include the JNK pathway, TGF-βsignaling, germline signaling, and mitochondrial respiration as well as other factors leading to aging process such as protein homeostasis, temperature, transcription factors, and so on. The representative genes that are involved in those pathways are listed in Figure 1.

Currently, *daf-16* is the most central aging related gene and is the downstream target of some of the pathways, most prominently, the IIS pathway. Its activity involves direct interaction with other genes, modulating the nuclear translocation or acting as a transcription factor to numerous target genes for lifespan regulation and stress resistance [43,44,45,46]. Not surprisingly, the pathways or factors that are involved in the aging process interact with each other to mediate lifespan extension.

## 4. Bioactive Phytochemicals with Health Benefits

Derived from natural sources such as plants, animals, and microorganisms, bioactive compounds exhibit a highly diverse chemical nature. They are capable of interacting with biochemical systems, thereby conferring a wide spectrum of health benefits such as anti-inflammatory, anti-aging, anti-hypertensive, anti-cancer, anti-diabetic, and anti-neurodegenerative properties [47,48,49,50,51,52,53]. Of all of the bioactive compounds, the phytochemicals from plants are the most prominent and have gained more attention since increasing numbers of lifespan-extending candidates have been discovered from phenotypic screening in *C. elegans* [54]. Reverse genetic screening along with structural studies has elucidated the targets of some of these compounds and the related pathways that are involved. Phytochemical/secondary metabolites can be found in vegetables, fruits, cereals, and inedible plants, and many of them have been identified to possess antioxidant properties [55]. They are chemically categorized into polyphenols, terpenoids, alkaloids, saponins, phytosterols, and organosulfur compounds [55,56,57,58,59]. Here we focused on the application of the most abundant and most chemically diverse phytochemicals (polyphenols, terpenoids, and alkaloids) to aging studies and further highlighted the use of crude extract in aging studies in the subsequent section.

### 4.1. Polyphenolic Compounds

The polyphenolic compounds include flavonoids, tannins, stilbenes, coumarins, lignans, and other phenolic compounds (Table 1) that are mainly involved in curbing oxidative stress and related conditions by providing their reducing power to protect essential cellular components from detrimental oxidative damage [60]. Among the members of this class, the flavonoids are the most abundant group, with over 8000 compounds having been identified [61]. Flavonoids contain a basic flavan nucleus with 15 carbon atoms that are grouped into C_6_-C_3_-C_6_ skeleton that comprises two aromatic C_6_ rings and a heterocyclic ring with one oxygen atom [62]. The presence of a highly reactive hydroxyl group enables flavonoids to donate the hydrogen atom, thereby reducing highly oxidizing free radicals [62]. The main targeted pathway linked to the anti-aging efficacy by this class of compounds is IIS. For example, Tambulin, as a hydroxy substituted flavanol, enhances stress tolerance and longevity and mitigates the manifestation of Parkinson-like symptoms in *C. elegans* model, during which the IIS pathway is up-regulated by the expressions of *daf-16*, *sod-1*, *sod-3*, and *ctl-2* [22]. Rosmarinic acid (RA) as a natural polyphenol, has been shown to improve the mean lifespan of *C. elegans* [63] by up-regulating the IIS pathway via *ins-18* and *daf-16*; the MAPK pathway via *skn-1* and *sek-1*; and the stress resistance and antioxidant genes such as *ctl-1*, *sod-3* and *sod-5*. The lifespan of the worms can also be extended by curcumin [64], which depends on the functions of *age-1*, *skn-1*, *sir-2.1*, *sek-1*, *unc-43*, *osr-1*, and *mek-1,* which are related to the IIS, MAPKK, and JNK signaling pathways. Polyphenol and chlorogenic acid enhance longevity via the IIS pathway depending on *daf-16*, *skn-1* and *hsf-1* [65]. Epigallocatechin gallate and epicatechin (EC) protect and facilitate the expression of major genes of the IIS pathway to protect *C. elegans* from oxidative stress, thereby enhancing the longevity [66,67,68]. Furthermore, pro-longevity effects have been found in the *C. elegans* aging models, with a broad variety of polyphenols/flavonoids such as myricetin, resveratrol, quercetin, naringenin, kaempferol, catechin, baicalein, fisetin, caffeic acid, phenethylester, acacetin, and blueberry polyphenols, most of which require *daf-16* [42,69,70,71,72,73,74,75,76,77,78,79,80].

### 4.2. Terpenoids

Terpenoids are the most abundant and structurally diverse phytochemicals. Also known as terpenes, they can be classified based on the number of isoprene units (C_5_H_8_)_n_ into various subgroups, including hemiterpenes (C_5_H_8_), monoterpenes (C_10_H_16_), sesquiterpenes (C_15_H_24_), diterpenes (C_20_H_32_), sesterterpenes (C_25_H_40_), triterpenes (C_30_H_48_), tetraterpenes (C_40_H_64_), and polyterpenes (C_5_H_8_)_n_. Terpenoids exhibit a broad range of pharmacological activities, ranging from anti-malarial, anti-cancer, anti-inflammatory, anti-bacterial, anti-viral, and anti-aging to anti-neurodegeneration [108]. Some of the notable terpenoids are the chemotherapy medication Taxol^®^ (paclitaxel) and the frontline antimalarial drug Artemisinin [108].

Isoprenol is a hemiterpene-based unsaturated C_5_ alcohol that confers lifespan extension and stress tolerance in *C. elegans* via *daf-16* and *skn-1* through the IIS pathway [23]. Moreover, the expressions of SOD-3 and GST-4 can be boosted by isoprenol through the translocation of DAF-16 from the cytosol into the nucleus [22]. Carnosic acid relies on the MAPK and HSF-1 pathway to up-regulate *sod-5*, *hsp-16.2*, *hsp-16.1*, *sek-1*, and *skn-1* [109]. Carnosol, as a phenolic diterpene, increases the mean lifespan of *C. elegans* through the HSF-1 signaling pathway to up-regulate *sod-3* and *sod-5* as antioxidants and *hsp-16.1* and *hsp-16.2* for heat shock response [110]. Beta-caryophyllene (BCP), a naturally occurring bicyclic sesquiterpene, is capable of extending lifespan as well as of increasing the resistance to oxidative stress by inducing the dietary restriction response and xenobiotic stress response [111]. The compound 4-Hydroxy-E-globularinin, an iridoid isolated from *Premma integrifolia*, exhibits detoxification activity against ROS and up-regulates *hsp-16.2* and *sod-3* through *daf-16* in the IIS pathway [112]. A similar effect was found with oleanolic acid, where *sod-3*, *hsp-16.2*, and *ctl-1* were up-regulated via *daf-16* [113]. However, α-Tocopherol, either in free form or encapsulated with SDNF (soluble dietary fiber-based nanofibers), may provide the longevity benefits via different routes [114]. Lifespan extension potentials have also been ascribed to additional high molecular weight isoprenoid-based compounds such as withanolide-A, specioside, ursolic acid, and glycyrrhetinic acid [60,115,116,117]. Examples of these terpenoids with longevity-modulating effects are summarized in Table 2.

### 4.3. Alkaloids

Alkaloids represent a class of nitrogenous chemicals that are not only derived from plants but also from fungi, bacteria, and animals [124]. According to their heterocyclic ring system and biosynthetic precursor, this class of compounds are categorized into eight subgroups, including tropanes, indoles, imidazoles, piperidines, isoquinolines, pyrrolizidines, quinolozidines, and pyrrolidine alkaloids [124]. Historically, some alkaloids have been used as poisons, whereas others have been used as remedies against fever and snakebites [124,125,126]. Though a bit under the shadow of toxicity, alkaloids elicit great potential in pharmaceutical development, primarily as analgesic, antioxidant, anti-inflammatory, anti-bacterial, anti-spasmodic, anti-cancer, anti-hypertensive, and stimulants to the central nervous system [56,124,127]. Particularly, over 300 compounds of this class have been shown to possess some degree of anti-aging property [40]. Reserpine confers significant thermo tolerance and longevity benefits to the worms, which is likely independent of *daf-16* and partially rely on serotonin [126]. Enhanced longevity effects are also offered by the methylxanthine alkaloid caffeine, which is able to induce the nuclear translocation of *daf-16* but does not requires its activity [128]. The action of pentagalloyl glucose (a gallotannin) on lifespan requires the cooperation between four pathways, including the IIS pathway, the mitochondrial ETC, the Sir-2.1 signaling, and the dietary restriction pathway [53]. The studied alkaloids with lifespan extending potentials in *C. elegans* are summarized in Table 3.

### 4.4. Plants Crude Drugs and Extracts with Lifespan Extending Abilities in C. elegans

Medicinal plants are essential sources of bioactive therapeutic compounds. Therapeutic studies using pure isolated phytochemicals are commonly preceded by preliminary studies using crude plant extracts to confirm the therapeutic potentials. Several studies have reported that some plant extracts show a lifespan prolonging effect on *C. elegans*. For instance, the aqueous stem bark extract from *Endopleura uchi* was reported to alter the DAF-16/FOXO pathway and to enhance the expression of the stress response genes such as *hsp-16.2* and *sod-3* [135]. Through High Performance Liquid Chromatography/Ultraviolet-Visible (HPLC UV/VIS) analysis, phenolic bergenin has been proposed as the major active component, potentially paving a new path for further developing compounds with similar and even more potent effects [135]. *Calycophyllum spruceanum* water extract has been shown to modulate the DAF-16/FOXO pathway, and five secondary metabolites have been identified via HPLC/Mass Spectrometry (MS) analyses, including 5-hydroxy-6-methoxycoumarin-7-glucoside, cyanidin, gardenoside, taxifolin, and 5-hydroxymorin [136]. Both the leaf and fruit extract of *Caesalpinia mimosoides* and *Eugenia uniflora* possess longevity enhancing activity via the IIS pathway with *sod-3*, *gst-4,* and *hsp-16.2* as targets [137,138]. *Glochidion zeylanicum* and *Anacardium occidentale* leaf extracts have also been reported to display longevity enhancing and oxidative stress resistance activities in *C. elegans* via the DAF-16/FOXO and SKN-1/Nrf-2 signaling pathways [139,140]. Furthermore, phytochemical analysis narrowed down on the active compounds, revealing benzoic acid, pentadecanoic acid, octadecatrienoic acid, n-hexadecanoic acid, β-caryophyllene, palmitic acid, and α-linolenic acid as prominent metabolites. Extract from *Hibiscus sabdariffa L.* exhibits significant lifespan extension activities alongside curbing amyloid-β toxicity in *C. elegans*, which is mediated by the IIS pathway through the activation of the DAF-16 and SKN-1 transcription factors [141]. An ayurvedic polyherbal extract (PHE) derived from six herbs, including *Berberis aristata*, *Emblica officinalis*, *Cyperus rotundus*, *Terminalia chebula Cedrus deodara,* and *Terminalia bellirica,* has been shown to enhance the expressions of *daf-16*, *daf-2*, *skn-1*, *sod-3,* and *gst-4*, all of which are associated with longevity and stress response [142]. As mentioned in the foregoing, although crude extracts may exert therapeutic effects, the main ingredients that are responsible are pure chemical components that serve specific effects. It is therefore important to elucidate and characterize the specific components, as this will eventually pave the way for new strategic pharmacological designs. Examples of these plant extracts with longevity-modulating effects are summarized in Table 4.

## 5. Summary and Perspectives

Globally, the significant rise in the aging populace is imposing a great economic and social burden. It is necessary to conduct more research focusing on the biological process of aging, with the aim of facilitating the development of potential interventions to alleviate the adverse health impact of aging-associated medical conditions such as neurodegenerative disorders, cancer, diabetes, and cardiovascular diseases. Suitable model organisms such as *C. elegans* will continuously provide more unique insight into this complex process as well as in the discovery of new bioactive compounds with pharmaceutical efficacy. Plant polyphenols including flavonoids, tannins, stilbene, coumarins, lignan, and other phenolic compounds as well as terpenoids and alkaloids have been a major source of these bioactive compounds, either as single purified compound or in crude extract mixtures. The disadvantage of crude extracts is that a complex interaction between all of the components of the crude and the biological system of the model may produce inhibitory, antagonistic, or synergistic effects. This makes it impossible to know exactly which component is inducing the particular effect being observed. Despite this challenge, crude the extracts of medicinal plants such as *Endopleura uchi*, *Calycophyllum spruceanum*, *Caesalpinia mimosoides*, *Eugenia uniflora*, *Glochidion zeylanicum*, *Anacardium occidentale*, *Hibiscus sabdariffa L,* and an ayurvedic polyherbal extract (PHE) derived from six herbs has demonstrated efficacy in extending lifespan in *C. elegans* [135,136,137,138,140,141,142]. However, purified compounds are better alternatives, as they provide the advantages of elucidating the structure, targets, mechanism of action, and pathways involved as well as the modification of the compound for improved activity.

In the future, discovering novel and effective natural candidates that extend lifespan and that delay aging and related diseases still depends on advancements in state-of-the-art high-throughput screening techniques [145,146]. Combining this effort with studies in suitable model organisms such as *C. elegans* [147,148] will provide a better platform for understanding the aging process as well as facilitating healthy aging to improve the quality of life of the elderly population.

## Figures and Tables

**Figure 1 molecules-26-07323-f001:**
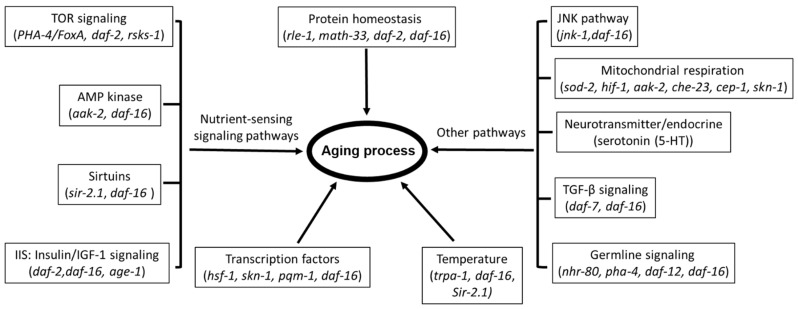
The known signaling pathways and key factors linked to aging process.

**Table 1 molecules-26-07323-t001:** Polyphenolic compounds with anti-aging and lifespan extending properties as demonstrated in the *C. elegans* model.

Polyphenolic Compounds	Mean Lifespan Extension *	Plant Source #	Ethnobotanical Use	Implicated * Genes	Pathway *	Anti-Aging Effect *	References
Flavonoids							
Tambulin	16% at 50 µM	*Zanthoxylum armatum* (Indian thorny ash, or Nepali Dhania, or Chinese coriander or timur)	Medicinal	*daf-16*, *sod-1*, *sod-3*, *ctl-2*	IIS	Anti-Parkinson’s	[22]
Rosmarinic acid	40% at 60 µM 49% at 120 µM 63% at 180 µM	*Rosmarinus officinalis* (Rosemary)	Medicinal, Ornamental, Culinary, Source of essential oil	*ins-18*, *daf-16*, *sek-1*, *skn-1*, *ctl-1*, *sod-3*, *sod-5.*	IIS and MAPK	Anti-oxidative, Healthspan extension	[63]
Curcumin	39% at 20 µΜ	*Curcuma longa* (Turmeric)	Medicinal, Culinary	*age-1*, *skn-1*, *sir-2.1*, *sek-1*, *unc-43*, *osr-1*, *mek-1**genes*	IIS, MAPK and JNK	Anti-oxidative, Healthspan extension	[64]
Chlorogenic acid	20% at 50 µM	*Coffea arabica* (Coffee), *Camellia sinensis* (Tea)	Medicinal, Beverage	*daf-16*, *skn-1* and *hsf-1*	IIS	Anti-oxidative, Healthspan extension	[65]
Epigallocatechin gallate (EGCG)	10%–14% at 220 µΜ	*Camellia sinensis* (Green tea)	Medicinal, Beverage	*daf-16*	IIS	Anti-oxidative, Anti-Alzheimer’s	[66,67]
	20% at 100 μM			*daf-16*, *sod-3*	IIS		[81]
Myricetin	32% at 100 µM	*Abelmoschus moschatus* (musk mallow), *Citrus sinensis* (Navel oranges), *Vaccinium sect. Cyanococcus* (Blueberry leaves)	Medicinal, Food	*daf-16*	IIS	Anti-oxidative	[69]
	18% at 100 µM			*daf-16*, *sod-3*	IIS		[74]
Quercetin	15% at 100 µM	*Allium cepa L.* (Onions), *Malus domestica* (Apples), *Brassica oleracea* (Broccoli), *Vitis vinifera* (Grape)	Functional food, Culinary	*daf-16*	IIS	Anti-oxidative, Healthspan extension	[73]
	5% at 100 µM			*daf-16*	IIS		[74]
	11% at 100 µM 18% at 200 µM			*age-1*, *daf-2*, *sek-1* and *unc-43*	IIS, CaMKII and p38 MAPK		[82]
Kaempferol	5% at 100 µM	*Camellia sinensis* (Tea), *Brassica oleracea* (Broccoli), *Vitis vinifera* (Grape), *Solanum lycopersicum* (Tomato), *Fragaria ananassa* (Strawberries), *Malus domestica* (Apples)	Beverage, Functional foods	*daf-16*	IIS	Anti-oxidative, Healthspan extension	[74]
	10% at 100 µM			*daf-16*	IIS		[77]
Fisetin	6% at 100 µM	*Fragaria ananassa* (Strawberries), *Malus domestica* (Apples), *Diospyros kaki* (Persimmons), *Allium cepa L.* (Onions), *Cucumis sativus* (Cucumbers)	Functional foods	*daf-16*	IIS	Anti-oxidative	[77]
Catechin	8% at 200 µM	*Camellia sinensis* (Green tea), *Theobroma cacao* (Cocoa), *Vitis vinifera* (Grape), *Malus domestica* (Apples)	Medicinal, Functional foods	*akt-2*, *mev-1*, *nhr-8*	IIS	Anti-oxidative, Healthspan extension	[75]
Epicatechin (EC)	15% at 100 µM	*Camellia sinensis* (Tea), *Theobroma cacao* (Cocoa), *Vitis vinifera* (Grape red wine)	Functional foods, Beverage	*daf-16*, *sod-3*	IIS	Anti-oxidative, Healthspan extension	[81]
	47% at 200 µM			*daf-2*, *age-1*, *akt-1*, *akt-2*, *sgk-1*, *daf-16*, *skn-1*, *hsf-1*, *gst-4*, *gst-7*, *hsp-16.2* and *hsp-70*	IIS		[68]
3′-O-methylepicatechin	6% at 200 µM	*Malus domestica* (Apples), *Vitis vinifera* (Grape), *Theobroma cacao* (Cocoa), *Camellia sinensis* (Tea)	Functional foods, Beverage	-	-	Anti-oxidative, Healthspan extension	[83]
4′-O-methylepicatechin	12% at 200 µM			-	-		[83]
Baicalein	45% at 100 µM	*Scutellaria baicalensis* (Baikal skullcap or Chinese skullcap)	Medicinal	*skn-1*	IIS	Anti-oxidative	[76]
	36% at 0.1% *w*/*v*			*cbp-1*	-		[84]
Caffeic acid	11% at 300 µM	*Coffea arabica* (Coffee)	Beverage, Medicinal	*osr-1*, *sek-1*, *sir-2.1*, *unc-43*, and *daf-16*		Anti-oxidative,	[85]
Acacetin (5,7-dihydroxy-4-methoxyflavone)	27% at 25 µM	*Tephroseris kirilowii* (Dog Tongue grass, Cotton bat)	Medicinal	*sod-3*, *gst-4*, *ctl-1* and *hsp-16.2*	IIS	Anti-oxidative, Stress Resistance	[86]
Acacetin 7-O-α-l-rhamnopyranosyl(1–2)β-D-xylopyranoside	39% at 25 µM	*Premna integrifolia* (Wind killer)	Medicinal	*sir-2.1*, *skn-1*, *daf-16*, and *hsf-1*		Anti-oxidative, Stress Resistance	[87]
Quercetin-3-O-dirhamnoside	21% at 200 µM	*Curcuma longa* (turmeric)	Culinary, Medicinal	-	-	Anti-oxidative, Stress Resistance	[39]
Quercetin-3-O-glucoside	23% at 25 µM	*Erica multiflora* (Winter Heather)	Medicinal	-	-	Anti-oxidative	[88]
Isorhamnetin (Quercetin 3′-O-methylether)	16% at 200 μM	*Ginkgo biloba* (ginkgo or gingko also known as the maidenhair tree), *Hippophae rhamnoides* (Sea buckthorn), *Vaccinium sect. Cyanococcus* (Blueberry)	Medicinal, Food	-	-	Anti-oxidative, Stress Resistance	[89]
Tamarixetin (Quercetin 4′-O-methylether)	11% at 200 μM	*Cyperus teneriffae* (Coco-grass)	Medicinal	-	-	Stress resistance, Anti-oxidative	[89]
Icariin	20% at 45 μM	*Herba epimedii* (Horny Goat Weed)	Medicinal	*daf-2*, *daf-16*, *hsf-1*	IIS	Anti-oxidative, Anti-neurodegenerative diseases	[90]
Icariside II	20% at 20 μM	*Herba* epimedii (Horny Goat Weed)	Medicinal	*daf-2*, *daf-16*, *hsf-1*	IIS	Anti-oxidative, Stress resistance, Anti-neurodegenerative diseases	[90]
Isoxanthohumol	2% at 100 μM	*Humulus lupulus* (hops)	Medicinal Beverage	*daf-16*	IIS	Anti-oxidative, Stress resistance	[91]
Silymarin	10% at 25 µM 24% at 50 µM	*Silybum marianum* (Milk thistle)	Medicinal	*-*	-	Anti-oxidative, Stress resistance, Anti- Alzheimer’s	[1]
Genistein	27% at 100 µM	*Vigna angularis* (adzuki bean)	Medicinal, Food	*hsp 16.2*, *sod-3*		Healthspan extension, Stress resistance	[92]
Taxifolin	51% at 820 µM	*Silybum marianum* (blessed thistle or milk thistle), *Carduus marianus* (Marian thistle or Our-Lady’s-thistle), *Allium cepa L.* (Onions)	Medicinal, Culinary	-	-	Stress Resistance, Ameliorates Cerebral Amyloid Angiopathy (ACAA)	[93]
Trolox (6-Hydroxy-2,5,7,8-tetramethylchroman-2-carboxylic acid	31% at 0.6 mM–3 mM	*Fragaria ananassa* (Strawberries)	Functional food	-	-	Anti-oxidative	[93]
Chicoric Acid	20% at 100µM	*Cichorium intybus* (Chicory), *Echinacea angustifolia* (Purple cone flower), *Lactuca sativa* (Lettuce), *Ocimum basilicum* (Basil)	Medicinal, Culinary	-	AMPK	Anti-oxidative	[94]
Naringin	23% at 50 µM	*Citrus grandis* (Pomelo), *Citrus paradise* (grapefruit), and *Citrus aurantium* (Bitter orange)	Functional food	*daf-16*, *daf-2*, *akt-1*, *akt-2. eat-2*, *sir-2.1*, *rsks-1,* and *clk-1*	IIS	Anti-oxidative, Anti-Alzheimer’s, Anti-Parkinson’s	[35]
**Tannins**							
Tannic acid	18% at 100 μM	*Camellia sinensis* (Tea), *Vitis vinifera* (Grape), *Arachis hypogaea* (Pea nuts)	Functional food	*sek-1*	MAPK	Anti-oxidative, Anti-Alzheimers’, Neuroprotective Others Anti-amyloidogenic, Antimicrobial, Anticancer, Antimutagenic	[95]
	24% at 0.01% *w*/*v*			*daf-16*	IIS		[84]
Pentagalloyl Glucose	18% at 160 μM	*Eucalyptus leaves* (Southern blue gum or blue gum)		*daf-16*, *age-1*, *eat-2*, *sir-2.1,* and *isp-1*	IIS, DR, SIR-2.1 and METC.	Anti-oxidative Others Estrogenic, Anti-inflammatory, Anti-oxidative, Anticancer	[53]
**Stilbene**							
Resveratrol	Variable effects at 100 μM	*Vitis vinifera* (Grape), Peanuts, *Theobroma cacao* (Cocoa), *Vaccinium sect. Cyanococcus* (Blueberry), *Vaccinium myrtillus* (Bilberry), *Vaccinium macrocarpon* (cranberry)	Functional food	*sir 2.1*	-	Anti-oxidative Others Antiviral, Anti-depressant, Anti-nociceptive, Anti-diabetic activities	[70]
	3% at 5 μM			-	-		[71]
	11% at 100 µg/mL				-		[72]
OxyResveratrol	31% at 1000 µM	*Morus alba* (white mulberry)	Functional food, Medicine	*sir-2.1*, *aak*	AMPK and SIR-2.1	Anti-oxidative, Neuroprotective	[96]
TSG (2,3,5,4′-Tetrahydroxystilbene-2-O-β-D-glucoside)	23% at 100 μM	*Polygonum multiflorum* (Tuber fleeceflower)	Medicinal	-	-	Anti-oxidative, Stress resistance	[97]
Polydatin	30% at 1 mM	*Vitis vinifera* (Grape)	Functional food	*sir-2.1*, *skn-1*, *sod-3*, and *daf-16*	IIS	Anti-oxidative, Stress resistance, Neuroprotective	[98]
Piceatannol	18% at 50 and 100 µM	*Passiflora edulis* (Passion fruit), *Camellia sinensis* (White Tea), *Vitis vinifera* (Grape)	Functional food	*daf-16*, *hsp 16.2*, *sod-3*, *sir-2.1*	IIS	Anti-oxidative Others Estrogenic, Anti-inflammatory, Anti-oxidative, Anticancer	[99]
**Coumarins**							
**^** Urolithin A (UA)	45% at 50 µM	*Vaccinium sect. Cyanococcus* (Blueberry), *Fragaria ananassa* (Strawberries), *Arachis hypogaea* (Pea nuts), *Quercus spp* (acorns), *Punica granatum* (pomegranates), *Juglans regia* (Walnut), *Rubus idaeus* (raspberries)	Functional food	-	-	Anti-oxidative, Healthspan extension	[100]
**^** Urolithin B (UB)	36% at 50 µM	*Vaccinium sect. Cyanococcus* (Blueberry), *Fragaria ananassa* (Strawberries), *Arachis hypogaea* (Pea nuts), *Quercus spp* (acorns), *Punica granatum* (pomegranates), *Juglans regia* (Walnut), *Rubus idaeus* (raspberries)	Functional food	-	-	Anti-oxidative, Healthspan extension	[100]
**^** Urolithin C (UC)	36% at 50 µM	*Vaccinium sect. Cyanococcus* (Blueberry), *Fragaria ananassa* (Strawberries), *Arachis hypogaea* (Pea nuts), *Quercus spp* (acorns), *Punica granatum* (pomegranates), *Juglans regia* (Walnut), *Rubus idaeus* (raspberries)	Functional food	-	-	Anti-oxidative, Healthspan extension	[100]
**^** Urolithin D (UD)	19% at 50 µM	*Vaccinium sect. Cyanococcus* (Blueberry), *Fragaria ananassa* (Strawberries), *Arachis hypogaea* (Pea nuts), *Quercus spp* (acorns), *Punica granatum* (pomegranates), *Juglans regia* (Walnut), *Rubus idaeus* (raspberries)	Functional food	-	*-*	Anti-oxidative, Healthspan extension	[100]
**Lignan**							
Sesamin	13% at 6.3 µg/plate	*Sesamum indicum L.* (Sesame Seeds)	Functional Food	*daf-2*, *skn-1*, *pmk-1*, and *daf-16*	IIS	Anti-oxidative Others Anti-allergenic, Anti-carcinogenic, Antihypertensive, Hypocholesterolemic	[101]
Vitexin	17% at 100 mM	*Vigna angularis* (adzuki beans)	Functional Food	*sod-3*, *hsp-16.2*	IIS	Anti-oxidative Others Antiviral, Anti-depressant, Anti-nociceptive, Anti-diabetic	[13]
Arctigenin	13% at 100 µM	*Arctium lappa* (Greater burdock)	Medicinal	*daf-16*, *jnk-1*	IIS	Anti-oxidative, Stress resistance	[102]
Matairesinol	25% at 100 µM	*Arctium lappa* (Greater burdock)	Medicinal	*daf-16*, *jnk-1*	IIS	Anti-oxidative, Stress resistance	[102]
Arctiin	15% at 100 µM	*Arctium lappa* (Greater burdock)	Medicinal	*daf-16*, *jnk-1*	IIS	Anti-oxidative, Stress resistance	[102]
Lappaol C	11% at 100 µM	*Arctium lappa* (Greater burdock)	Medicinal	*daf-16*, *jnk-1*	IIS	Anti-oxidative, Stress resistance	[102]
Lappaol F	12% at 100 µM	*Arctium lappa* (Greater burdock)	Medicinal	*daf-16*, *jnk-1*	IIS	Anti-oxidative, Stress resistance	[102]
(Iso) lappaol A	11% at 100 µM	*Arctium lappa* (Greater burdock)	Medicinal	*daf-16*, *jnk-1*	IIS	Anti-oxidative, Stress resistance	[102]
**Phenolic Compounds**							
Tryosol	21% at 250 μM	*Olea Europea L.* (Olive tree)	Medicinal, Food	*hsf-1*, *daf-2*, *daf-16*	IIS	Anti-oxidative, Stress resistance	[103]
6-Gingerol	20% at 25 µM	*Zingiber officinale* (Ginger)	Culinary, Medicinal	*hsp-16.2*, *sod-3*		Anti-oxidative, Stress resistance	[5]
6-Shogaol	19% at 12.5 µM 25% at 25 µM	*Zingiber officinale* (Ginger)	Culinary, Medicinal	*sod-3*, *hsp-16.2*		Anti-oxidative, Stress resistance	[5]
Salicyclic Acid	14% at 1 mM	*Rubus idaeus* (raspberries), Salix alba L. (Willow tree)	Medicinal, Food	*daf-16*, *sod-3*, *sod-5*, *ctl-2*, *gst-4*, *gst-10*	IIS	Anti-oxidative, Stress resistance	[104]
Salicylamine	32% at 100 µM 56% at 500 µM	*Fagopyrum esculentum* (Buckwheat)	Culinary	*sir-2.1*, *ets-7*	-	Healthspan extension	[105]
Juglone	29% at 40 µM	*Juglans nigra* (Black Walnut)	Functional Food, Medicinal	*daf-16*, *sod-3*, *hsp-16.2*, *sir-2.1*	IIS	Anti-oxidative, Stress resistance	[106]
Gallic Acid	25% at 300 μM	*Punica granatum* (Pomegranate), *Aspalathus linearis* (Rooibos tea), *Vitis vinifera* (Grape), *Raphanus sativus* (Black radish), *Allium cepa L.* (Onions)	Functional food	-	-	Stress resistance	[17]
Ferulic Acid	9.58% at 500 µM	*Beta vulgaris* (Beet root), *Oryza sativa* (Rice), *Glycine max* (Soyabean), *Daucus carota* (Carrot), *Avena sativa* (Oats)	Functional food	*daf-2*, *daf-16*, *hlh-30*, *skn-1*, and *hsf-1*	IIS	Healthspan extension, Stress resistance, Anti- Huntington’s disease	[107]

* Mean lifespan and implicated genes, pathways, and anti-aging effects in the table are the specific outcomes from the references indexed directly against the compound at the right-end column. A dash (-) sign in the column for implicated genes and pathways represents where the authors did not proceed to investigate the molecular mechanism beyond the effect observed. Where different mean lifespans are not from the same study, we have referenced the appropriate literature for such. Original units of concentration used by the investigators are reported here. Additionally, where other health benefits besides the anti-aging effect have been noted for the compound in the primary research referenced, we grouped them under the heading “Others” in the column for “Anti-aging-effect”. **#** Plant sources of the compound in the study indexed to the reference directly at the right-end column are listed. Other sources that have been mentioned by the primary study, and previous studies have also been listed as well. Scientific names and common names (in parenthesis) of the plant sources are provided. **^** Compounds not directly obtained from the plant sources indicated but that are produced by gut microflora from foods rich in ellagitannins such as the plants listed.

**Table 2 molecules-26-07323-t002:** Terpenoids with anti-aging and lifespan extending properties as demonstrated in the *C. elegans* model.

Terpenoids	Mean Lifespan Extension *	Plant Source #	Ethnobotanical Use	Implicated * Genes	Pathway *	Anti-Aging Effect *	References
Carnosic Acid	3% at 60 µM 8% at 120 µM 16% at 180 µM	*Rosmarinus officinalis L* (Rosemary)	Food, Medicinal	*sod-5*, *hsp-16.2*, *hsp-16.1*, *sek-1*, *skn-1*	MAPK and HSF-1	Anti-oxidant, Anti-inflammatory, Antibacterial, Anti-cancer, Neuroprotective	[109]
Carnosol	19% at 180 µM	*Rosmarinus officinalis L* (Rosemary)	Medicinal, Culinary	*sod-3*, *sod-5*, *hsp-16.1*, *hsp-16.2*, *hsf-1*, *daf-16.*	IIS	Antioxidant, Anticancer, Antimicrobial, Anti-inflammatory	[110]
Beta-Caryophyllene	>22% at 50 μM	*Syzygium aromaticum* (Clove), Cannabis sativa (hemp), *Rosmarinus officinalis L* (Rosemary, Humulus lupulus (Hops)	Culinary, Beverages	*pha-4*, *sir-2.1*, *hsf-1*, *skn-1*, *daf-16*, *gst-4*, *gst-7*, *hsp-70*, *sod-2*, *sod-3* and *daf-9*	IIS	Anti-oxidant, Anti-inflammatory, anti-biotic, Anti-carcinogenic, local anesthetic	[111]
4-Hydroxy-E-globularinin	18% at 20 μM	*Premna integrifolia* (Wind killer)	Medicinal	*daf-16*, *hsp-16.2*, *sod-3*	IIS	Anti-oxidant	[112]
10-*O*-*trans*-*p*-Coumaroylcatalpol	17% at 20 μM	*Premna integrifolia* (Wind killer)	Medicinal	*daf-16*	IIS	Anti-oxidant, Anti-parkinson’s disease	[118]
Oleanolic acid	16% at 300 μM	Constituent of the leaves and roots of more than 120 plant species such as *Olea europaea* (olive tree), *Viscum album* (European mistletoe or common mistletoe), *Aralia chinensis* (Chinese Angelica Tree)	Food, Medicinal	*sod-3*, *hsp-16.2*, *ctl-1*, *daf-16*	IIS	Anti-oxidant, Hepatoprotective, Hypoglycemic, Anti-inflammatory	[113]
α-Tocopherol	7% at 50 µg/mL 15% at 100 µg/mL 17% at 200 µg/mL	Sunflower seeds (Helianthus annuus), *Prunus dulcis* (Almonds), *Corylus avellana L.* (Hazelnuts), *Arachis hypogaea* (Pea nuts), *Spinacia oleracea* (Spinach), *Brassica oleracea var. italica* (Broccoli), *Actinidia deliciosa* (Kiwifruit), *Mangifera indica* (Mango)	Functional food	-	-	Anti-oxidant	[114]
Withanolide-A	29% at 5 μM	*Withania Somnifera* (Ashwagandha)	Medicinal	*sgk-1*, *daf-16*, *sod-3*, *skn-1*, *hsf-1*, *gst-4*, *hsp-16.2*	IIS	Neuroprotective, Stress resistance	[115]
Specioside	15% at 25 μM	*Stereospermum suaveolens* (Patala)	Medicinal	*sod-1*, *sod-2*, *sod-3*, *gst-4*, *gst-7*, *hsp-16.2*, *hsp-70*, *clt-1*	IIS	Antioxidant, Stress resistance	[87]
Ursolic acid	32% at 25 μM	*Malus domestica* (Apple peels), , rosemary, *Lavandula angustifolia* (lavender), *Mentha piperita* (Peppermint), Thymus vulgaris (thyme), *Ocimum basilicum* (Basil), *Vaccinium myrtillus* (Bilberry)	Medicinal, Culinary	*jnk-1*, *jkk-1*	JNK-1	Antioxidant, Stress resistance	[117]
18α-Glycyrrhetinic acid	17% at 20 μg/mL	*Glycyrrhiza glabra* (Licorice)	Medicinal, Culinary	*skn-1*, *daf-16*	p38 MAPK	Neuroprotective	[59]
Glaucarubinone	1.9 days at 100 nM	*Simaroubaceae spp.* (Amargo, Bitterwood, Marupa, or Quassia)	Ornamental, Medicinal	-	-	Anti-oxidant Others Antimalarial	[71]
Fucoxanthin	14% at 5 μM	*Undaria pinnatifida* (Wakame), *Hijikia fusiformis* (Hijiki)	Medicinal	-	-	Antioxidant, Stress resistance	[119]
Catalpol	28% at 25 μM	*Rehmannia glutinosa* (Chinese foxglove)	Medicinal plant	*mek-1*, *daf-2*, *age-1*, *daf-16*, and *skn-1*	IIS	Anti-oxidant, Anti- Alzheimer’s, Anti-Parkinson’s, Anti-stroke Others Anticancer, Anti-diabetes	[120]
Ferulsinaic acid	20% at 100 µM	*Ferula* communis (Giant Fennel)	Medicinal, Culinary	-	-	Anti-oxidant	[121]
Verminoside	20% at 25 µM	*Stereospermum suaveolens* (Patala)	Medicinal	*daf-16*	-	Antioxidant, Stress resistance	[116]
Dehydroabietic acid	15% at 10 µM	*Pinus densiflora* (Japanese red pine), *Pinus sylvestris* (Scots pine), *Abies grandis*(Grand fir)	Medicinal	*sir-2.1*	-	Healthspan extension	[122]
Secoisolariciresinol Diglucoside	22% at 500 µM	*Linum usitatissimum* (*Flaxseed*)	Food, Medicine	*daf-16*, *hsf-1*, *nhr-80*, *daf-12*, *glp-1*, *eat-2*, and *aak-2.*	IIS	Anti-oxidant, Anti-Alzheimer’s, Anti-Parkinson’s	[123]

* Mean lifespan and implicated genes, pathways, and anti-aging effects in the table are the specific outcomes from the references indexed directly against the compound at the right-end column. A dash (-) sign in the column for implicated genes and pathways represents where authors did not proceed to investigate the molecular mechanism beyond the effect observed. Where different mean lifespans are not from the same study, we have referenced the appropriate literature for such. Original units of concentration used by the investigators are reported here. Additionally, where other health benefits besides the anti-aging effect have been noted for the compound in the primary research referenced, we grouped them under the heading “Others” in the column for “Anti-aging-effect”. # Plant sources of the compound in the study indexed to the reference directly at the right-end column are listed. Other sources that have been mentioned by the primary study and previous studies have also been listed as well. Scientific names and common names (in parenthesis) of the plant sources are provided.

**Table 3 molecules-26-07323-t003:** Alkaloids with anti-aging and lifespan extending properties as demonstrated in the *C. elegans* model.

Alkaloids	Mean Lifespan Extension *	Plant Source #	Ethnobotanical Use	Implicated * Genes	Pathway *	Anti-Aging Effect *	References
Reserpine	31% at 30 μM	*Rauwolfia* serpentine (Indian snakeroot), *Rauwolfia vomitoria* (the poison devil’s-pepper)	Medicinal	*tph-1* (*serotonin*)	Serotonin pathway	Anti-oxidant, Antipsychotic, Anti-hypertensive	[126]
Tomatidine	7% at 25 μM	*Solanum lycopersicum* (Unripe tomato fruits, leaves and stems)	Medicinal, Functional food	*skn-1*	IIS	Anti-inflammatory, Anti- tumorigenic, Lipid-lowering activities	[129]
Spermidine	18% at 0.2 mM	*Glycine max* (soy bean), *Pisum sativum* (green peas), *Zea mays* (Maize corn)	Functional food	-	-	Enhanced autophagy	[130]
	15% at 0.2 mM			-	-		[131]
Caffeine	29% at 0.1% *w*/*v*	*Theobroma cacao* (Cocoa beans), *Cola acuminata* (kola nuts), *Camellia sinensis* (Tea leaves), *Coffea arabica* (coffee beans)	Beverages, Medicinal	*daf-16*	IIS	Antioxidant, Stress resistance, Neuroprotective, Anti-Alzheimer’s	[84]
	16% at 10 mM			*daf-16*	IIS		[3]
	80% at 5 mM			*daf-2*	IIS		[128]
	31.9% at 5 mM			*skn-1*, *gst-4*	IIS		[127]
Theophylline	25% at 5 mM	*Camellia sinensis* (Tea), *Coffea arabica* (Coffea)	Beverages, Medicinal	*skn-1*, *gst-4*	IIS	Antioxidant, Stress resistance	[127]
Chlorophyll	23% at 10 µg/mL 25% at 40 µg/mL	*Spinacia oleracea* (Spinach)	Food, Medicinal	*daf-16*, *sod-3*	IIS	Antioxidant	[132]
Pyrroloquinoline quinone	33% at 0.5 mM	*Actinidia deliciosa* (Kiwifruit), *Petroselinum crispum* (Parsley), *Capsicum annuum* (Green bell pepper), *Carica papaya* (Pawpaw)	Functional food, Culinary	*daf-16,skn-1*, *sod-3*, *hsp 16.2*, *gst-1* and *gst-10*	IIS	Antioxidant, Stress resistance	[133]
Calycosin	21% at 200 µM	*Astragalus mongholicus* Bunge (membranous milk-vetch)	Medicinal	*daf-16*, *hsp-16.2*, *ctl-1*, *sod-3*	IIS	Antioxidant, Stress resistance	[134]

* Mean lifespan and implicated genes, pathways, and anti-aging effects in the table are the specific outcomes from the references indexed directly against the compound at the right-end column. A dash (-) sign in the column for implicated genes and pathways represents where authors did not proceed to investigate the molecular mechanism beyond the effect observed. Where different mean lifespans are not from the same study, we have referenced the appropriate literature for such. Original units of concentration used by the investigators are reported here. Additionally, where other health benefits besides the anti-aging effect have been noted for the compound in the primary research referenced, we grouped them under the heading “Others” in the column for “Anti-aging-effect”. # Plant sources of the compound in the study indexed to the reference directly at the right-end column are listed. Other sources that have been mentioned by the primary study and previous studies have also been listed as well. Scientific names and common names (in parenthesis) of the plant sources are provided.

**Table 4 molecules-26-07323-t004:** Plant crude extracts with anti-aging and lifespan-extending properties as demonstrated in the *C. elegans* model.

Plant Source #	Mean Lifespan Extension *	Ethnobotanical Use	Implicated Genes *	Pathway *	Anti-Aging Effect *	References
*Endopleura uchi* (*Uxi*)	33% at 300 µg/mL	Medicinal	*daf-16*, *hsp-16.2* and *sod-3*	IIS	Antioxidant, Stress resistance, Anti-Huntington’s disease	[135]
*Calycophyllum spruceanum* (*capirona*)	16% at 300 µg/mL	Medicinal	*daf-16*	IIS	Antioxidant, Stress resistance, Healthspan extension	[136]
*Caesalpinia mimosoides* (*Pansi*)	4% at 50 µg/mL	Food vegetable	*daf-16*, *sod-3*, *gst-4*	IIS	Antioxidant, Stress resistance	[137]
*Eugenia uniflora* (*Surinam cherry*)	Significant increase at 500 µg/mL	Food, Medicinal	*daf-16*, *hsp-16.2* and *sod-3*	IIS	Antioxidant, Stress resistance, Healthspan extension	[138]
*Anacardium occidentale* (*Cashew*)	20% by 50 μg/mL	Medicinal, Functional food	*daf-16*, *skn-1*, *sod-3*, *gst-4*	IIS	Antioxidant, Stress Resistance, Healthspan extension	[139]
*Glochidion zeylanicum* (*Umbrella Cheese*)	10% at 100 µg/mL	Medicinal, Food	*daf-16*, *skn-1*, *sod-3*, *gst-4*	IIS	Antioxidant, Stress Resistance, Healthspan extension	[140]
*Hibiscus sabdariffa L.* (*Roselle*)	24% at 1 mg/mL	Medicinal, Beverage, Food supplement	*daf-16*, *skn-1*	IIS	Antioxidant, Stress Resistance, Anti-Neurodegenerative	[141]
*Polyherbal extract of Berberis aristata* (*Indian barberry*); *Emblica officinalis* (*Indian gooseberry or amla*); *Cyperus rotundus* (*Purple Nutsedge*); *Terminalia chebula* (*gall nut*); *Cedrus deodara*(*Himalayan cedar*); *Terminalia bellirica* (*beleric myrobalan*)	16% at 0.01 µg/mL	Medicinal	*daf-16*, *daf-2*, *skn-1*, *sod-3* and *gst-4*	IIS	Antioxidant, Stress Resistance, Anti-Neurodegenerative	[142]
*Betula utilis* (Himalayan Silver Birch)	35.99 % at 50 μg/mL	Medicinal	*daf-16*, *hsf-1*, *skn-1*, *sod-3* and *gst-4.*	IIS	Antioxidant, Healthspan extension	[143]
*Citrus sinensis* (Orange extracts)	10.5%, 18.0%, and 26.2% at 100, 200, and 400 mg/mL, respectively	Functional food	*daf-16*, *sod-3*, *gst-4*, *sek-1*, and *skn-1*	IIS	Antioxidant, Healthspan extension	[35]
*Cuscuta chinensis* (Chinese Dodder)	24% at 30 µg/mL	Medicinal	*hsp-16.1* and *hsp-12.6*	IIS	Stress Resistance, Healthspan extension	[144]
*Eucommia ulmoides* (Hardy Rubber Tree)	9% at 30 µg/mL		-	-	Stress resistance	[144]

* Mean lifespan and implicated genes, pathways, and anti-aging effects in the table are the specific outcomes from the references indexed directly against the compound at the right-end column. A dash (-) sign in the column for implicated genes and pathways represents where authors did not proceed to investigate the molecular mechanism beyond the effect observed. Where different mean lifespans are not from the same study, we have referenced the appropriate literature for such. Original units of concentration used by the investigators are reported here. Additionally, where other health benefits besides the ant-aging effect have been noted for the compound in the primary research referenced, we grouped them under the heading “Others” in the column for “Anti-aging-effect”. # Plant sources of the compound in the study indexed to the reference directly at the right-end column are listed. Other sources that have been mentioned by the primary study and previous studies have also been listed along. Scientific names and common names (in parenthesis) of the plant sources are provided.

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
