# Peer review of "Bioactive Phytochemicals with Anti-Aging and Lifespan Extending Potentials in Caenorhabditis elegans"

_molecules, 2021, doi:10.3390/molecules26237323_

Round 1

Reviewer 1 Report

Comments are in the attached file. 

Reviewer 2 Report

Okoro and colleagues discussed the aging/longevity pathways, phytochemicals and their anti-aging properties in C. elegans as an aging model. This article is a well-organized review, deals with an interesting topic and contains relevant data. The followings are the general, technical and scientific comments:

  • In the whole text, please follow the rule of abbreviations in the text. For the first time, use both a full and abbreviated form, next time, just use an acronym (L129 (IIS), 169, 179, 180, …).
  • L126, “C. elegans should be italicized.
  • This manuscript has only one figure. I would suggest authors to draw more schematic figs to improve the quality of the manuscript.
  • It’s NOT acceptable to use tables to replicate the information already presented in the text please pay more attention to that.
  • Authors should format the references according to the journal instructions. Please pay attention to the title of articles, journal titles, etc. (Ref. No. 99, 102, 118,…).

Reviewer 3 Report

The manuscript entitled ‘Bioactive Phytochemicals with Anti-Aging and Lifespan Extending Potentials’ by Okoro  et al.,  required minor revision before its acceptance in Molecules. The Review is well written and suitable for publications after incorporating below comments

Authors please add Botanical name of the plants in tables to maintain uniformity. In some cases they added common name and in other botanical name.

Homogenize the use of terminology throughout the manuscript.

The paper need careful formatting in Tables while using secondary metabolities they Added Lignan and phenolics in capital letter, while other metabolities not like not. Maintain uniformity.

Also add literature on important medicinal plants such as Berberis lycium in combating inflammatory disorders as authors also mention a species of Berberis.

Add latest publications. The manuscript is not updated

Round 2

Reviewer 4 Report

The manuscript has been corrected and can be recommended for the publication.